# Alumina Ceramic Exacerbates the Inflammatory Disease by Activation of Macrophages and T Cells

**DOI:** 10.3390/ijms21197114

**Published:** 2020-09-26

**Authors:** Seong-Min Lim, Juyoung Hwang, Hae-Bin Park, Chan Ho Park, Jun-O Jin

**Affiliations:** 1Shanghai Public Health Clinical Center & Institutes of Biomedical Sciences, Shanghai Medical College, Fudan University, Shanghai 201508, China; i0473543@ynu.ac.kr (S.-M.L.); jyhwang5@yu.ac.kr (J.H.); ys02104@ynu.ac.kr (H.-B.P.); 2Department of Medical Biotechnology, Yeungnam University, Gyeongsan 38541, Korea; 3Research Institute of Cell Culture, Yeungnam University, Gyeongsan 38541, Korea; 4Department of Orthopedic Surgery, Yeungnam University Medical Center, 170, Hyeonchung-ro, Nam-gu, Daegu 42415, Korea

**Keywords:** Al_2_O_3_ FP, macrophage, CD4 T cell, CD8 T cell, inflammation

## Abstract

(1) Background: Aluminum oxide (Al_2_O_3_) ceramic is one of the materials used for artificial joints, and it has been known that their fine particles (FPs) are provided by the wear of the ceramic. Al_2_O_3_ FPs have been shown to induce macrophage activation in vitro; however, the inflammatory effect in vivo has not been studied. (2) Methods: We examined the in vivo effect of Al_2_O_3_ FPs on the innate and adaptive immune cells in the mice. (3) Results: Al_2_O_3_ FPs promoted the activation of spleen macrophages; however, conventional dendritic cells (cDCs), plasmacytoid DCs (pDCs), and natural killer (NK) cells were not activated. In addition, increases in the CD4 and CD8 T cells was induced in the spleens of the mice treated with Al_2_O_3_ FPs, which differentiated into interferon-gamma (IFN-γ)-producing helper T1 (Th1) and cytotoxic T1 (Tc1) cells. Finally, the injection of Al_2_O_3_ FPs exacerbated dextran sulfate sodium (DSS)-induced inflammation in the colon, mediated by activated and increased number of CD4 and CD8 T cells. (4) Conclusions: These data demonstrate that FPs of Al_2_O_3_ ceramic may contribute to the exacerbation of inflammatory diseases in the patients.

## 1. Introduction

Alumina ceramic articulation, made of aluminum oxide (Al_2_O_3_), is one of the bearing surfaces of hip arthroplasty, which is well known to have high resistance to wear and high chemical stability [1,2,3]. For these reasons, alumina ceramic articulation has been used for several decades.

The hip joint is the biggest joint of the body, which endures three times the body weight during walking. Joints perform tens of thousands of frictional movements in their lifetime [4]. Although the wear rate of alumina ceramic has been shown to be less than 1 μm/year, the fine particles (FPs) of Al_2_O_3_ are generated by continuous frictional movement, which may promote inflammation in the adjacent tissue of implant [2,3,5,6]. Previous studies have shown that the treatment of Al_2_O_3_ FPs induced activation of human macrophages, which produced higher levels of tumor necrosis factor-α (TNF-α) [7,8,9,10]. These inflammatory responses are capable of causing disastrous situation such as aseptic loosening [11]. However, the effects of Al_2_O_3_ FPs on the other immune cells, and in vivo effects in mouse models, have not been studied.

Inflammation is a reaction of the body’s defense system to tackle the pathogens [12,13]. Although the inflammation contributes in protecting the body by immune activation, it also induces damage to healthy tissues [12]. Various stimuli including pathogens, radiation, antigen, and damaged cells can promote inflammation, which is harmful to the healthy tissue [14,15,16]. Inflammatory responses are classified as acute and chronic reactions [17,18]. The acute inflammatory response is an initial response to the stimulus, which progresses within a short period of time, and is mainly involved in the innate immune cells such as macrophages, dendritic cells (DCs), and natural killer (NK) cells [19,20,21]. The DCs contain two main subsets, conventional DCs (cDCs) and plasmacytoid DCs (pDCs) [22]. The pDCs are the main cells that protect body from viral infection and produce interferon-alpha (IFN-α) in responses to the virus [23]. On the other hand, cDCs and macrophages are representative antigen-presenting cells (APCs) against microbes, which further induce the activation of T and B cells [24,25,26,27]. The activated DCs and macrophages upregulate co-stimulator, major histocompatibility complex, and pro-inflammatory cytokine production, which promote the differentiation and activation of T cells [28,29,30].

Inflammatory bowel disease (IBD) is a condition in which abnormal chronic inflammation (over 6 months) in the intestinal tract repeats improvement and recurrence [31,32]. A typical IBD is colitis and is caused by various reasons [33,34]. Dextran sulfate sodium (DSS) administration, which mediates helper T1 (Th1) and Th17 immune responses, is a common method for studying colitis [35,36,37,38].

Although it has been shown that the Al_2_O_3_ FPs induced the macrophage activation in vitro, the effects of Al_2_O_3_ FPs in the induction of inflammation in peripheral tissue in vivo in the mouse have not been studied. In addition, there was no report of the effect of Al_2_O_3_ FPs-induced inflammation in autoimmune disease. So, this study confirmed whether Al_2_O_3_ FPs exacerbates the inflammatory response by inducing IBD, the most representative autoimmune disease, into a mouse model. Therefore, we conducted this study to confirm our hypothesis that the Al_2_O_3_ FPs may induce the activation of innate and adaptive immune cells in vivo in the mice and exacerbate the tissue inflammation.

## 2. Results

### 2.1. Number of Spleen Macrophages Was Increased by Al_2_O_3_ FPs

Al_2_O_3_ FPs were suspended in distilled water and observed and analyzed under transmission electron microscopy (TEM). The Al_2_O_3_ FPs were mostly spherical (Figure 1a) and their size varied from 20 to 160 nm (Figure 1b).

Then, we evaluated the inflammatory properties of Al_2_O_3_ FPs in the mouse in vivo, especially on innate immune cells. The C57BL/6 mice were intraperitoneally (*i.p.*) injected with 10 mg/kg of Al_2_O_3_ FPs. The lipopolysaccharide (LPS) was also injected into mice as a positive control. Six hours post-injection, the spleen was harvested, revealing that the total number of splenocytes was not changed by Al_2_O_3_ FPs (Figure 2a). We further analyzed the activation of innate immune cells by Al_2_O_3_ FPs. As shown in Figure 2b, the pDCs (CD317^+^B220^+^ cells), cDCs (CD11c^+^lineage^-^ cells), macrophages (CD11b^+^CD11c^-^ cells), and NK cells (NK1.1^+^CD3^-^ cells) were defined in the spleen. We found that the number of macrophages was significantly increased by the treatment with Al_2_O_3_ FPs, whereas the number of NK cells tended to decrease (Figure 2c). The numbers of pDCs and cDCs were unchanged in the spleen compared to those in the phosphate buffered saline (PBS)-treated controls (Figure 2c). Thus, these data indicate that the Al_2_O_3_ FPs promote an increase in the number of macrophages in the spleen.

### 2.2. Al_2_O_3_ FPs Promoted the Activation of Spleen Macrophages

We further examined whether the Al_2_O_3_ FPs can induce the activation of innate immune cells. The treatment with 10 mg/kg of Al_2_O_3_ FPs did not upregulate the expression levels of co-stimulators and major histocompatibility complex (MHC) class I and II in pDCs and cDCs, although the CD86 expression levels were considerably increased in cDCs (Figure 3a,b and Appendix A). In contrast with DCs, the co-stimulator and MHC class I and II levels in the macrophages were significantly increased by the treatment with 10 mg/kg of Al_2_O_3_ FPs (Figure 3c and Appendix A). The expression of co-stimulators and MHC class were not considerably changed in pDCs, cDCs and macrophages at 1 and 5 mg/kg of Al_2_O_3_ FPs (Appendix A). Moreover, the co-stimulators were rapidly reduced 18 h after treatment of 10 mg/kg of Al_2_O_3_ FPs, whereas the MHC class I and II levels were significantly upregulated in the macrophages (Appendix A). On the other hand, the treatment with Al_2_O_3_ FPs did not promote the activation of NK cells in the spleen, although the CD69 expression levels were slightly increased (Appendix A).

We then examined the production of pro-inflammatory cytokines in the serum in response to Al_2_O_3_ FPs. The concentrations of tumor necrosis factor-alpha (TNF-α), interleukin (IL)-6, and IL-12p70 in the serum were substantially increased by Al_2_O_3_ FPs (Figure 3d). Thus, these data suggested that the Al_2_O_3_ FPs induce the activation of spleen macrophages, which contributes to the production of inflammatory cytokines in the serum.

### 2.3. Al_2_O_3_ FPs Elicited Helper T 1 (Th1) and Cytotoxic T 1 (Tc1) Cell Responses

Since the treatment with Al_2_O_3_ FPs promoted the activation of macrophages and production of pro-inflammatory cytokines, we examined whether the Al_2_O_3_ FPs induce activation and differentiation of T cells in adaptive immunity. C57BL/6 mice were *i.p.* injected with PBS, Al_2_O_3_ FPs, and LPS daily for 3 days, and their spleens were harvested on the day 4 as shown in the schematic diagram in Figure 4a. NK1.1^-^CD3^+^ cells in the live leukocytes were defined as T cells, which were further divided into CD4 and CD8 cells (Figure 4b). The numbers of total CD4 and CD8 T cells were significantly increased by the treatment with Al_2_O_3_ FPs (Figure 4c). Moreover, interferon-gamma (IFN-γ)-producing CD4 and CD8 T cells in the spleen were markedly increased by Al_2_O_3_ FPs, indicating that the Al_2_O_3_ FPs promote differentiation of Th1 and Tc1 cells (Figure 4d,e). In contrast with the interferon-gamma (IFN-γ) producing T cells, IL-4- and IL-17-producing CD4 T cells and IL-4-producing CD8 T cells were not changed considerably (Figure 4d,e). In addition, the concentration of IFN-γ in the serum was also elevated substantially by the treatment with Al_2_O_3_ FPs (Figure 4f). Thus, these data suggested that Al_2_O_3_ FPs elicit the Th1 and Tc1 responses in the mouse spleen.

### 2.4. DSS-Induced Colitis Was Exacerbated by Al_2_O_3_ FPs

Our data on the Al_2_O_3_ FPs-mediated induction of Th1 and T1c immune responses prompted us to examine whether the Al_2_O_3_ FPs exacerbate the progression of Th1-mediated inflammatory disease. The effects of Al_2_O_3_ FPs were examined in the DSS-induced colitis in the mice, where Th1 cells are the main contributors. C57BL/6 mice were randomly divided into four groups (five mice for each group) as shown in the schematic diagram (Figure 5a). The administration of DSS did not reduce the length of colon, whereas the additional treatment with Al_2_O_3_ FPs induced significant decrease in the length of colon compared to the controls (Figure 5b,c). Histological analysis showed that the DSS administration induced mild infiltration of leukocytes in the colon, which was substantially increased by the additional treatment with Al_2_O_3_ FPs (Figure 5d). Consistent with the histology data, the colon infiltrated number of leukocytes and the number of leukocytes in the mesenteric lymph node (mLN) was significantly increased by the treatment with DSS and Al_2_O_3_ FPs compared to those in the controls (Figure 5e). Therefore, these data suggested that the DSS-induced colitis was exacerbated by Al_2_O_3_ FPs in the mice.

### 2.5. Al_2_O_3_ FPs-Induced Th1 and Tc1 Cells Contributed in the Exacerbation of Colitis

We further examined whether the Th1 and Tc1 responses induced by Al_2_O_3_ FPs contributed in the exacerbation of DSS-induced colitis. Treatment with DSS and Al_2_O_3_ FPs elicited great increases in the number of total T cells in the colon and mLN as well as CD4 and CD8 T cells compared to those in controls (Figure 6a,b). Moreover, the TNF-α-, IFN-γ-, and IL-17A-producing CD4 T cells in the colon and mLN were significantly increased by Al_2_O_3_ FPs compared to those in controls (Figure 6c,d). Furthermore, IFN-γ-producing CD8 cells were also increased dramatically, whereas TNF-α-producing CD8 T cells were not increased compared to those in controls (Figure 6c,d). The serum concentration of IFN-γ was also elevated significantly upon treatment with DSS and Al_2_O_3_ FPs in the mice, whereas that of TNF-α was not increased upon treatment with DSS and Al_2_O_3_ FPs compared to treatment with the Al_2_O_3_ FPs alone and DSS alone (Appendix A). Thus, these data suggested that the Th1 and Tc1 immune responses induced by Al_2_O_3_ FPs contributed in the exacerbation of the DSS-induced colitis.

## 3. Discussion

Total hip arthroplasty has been proven to be an effective procedure to restore patients’ damaged joints by replacing them with the artificial joints [39]. Bearing surface is the most important part for the longevity of the arthroplasty. Although the bearing options have evolved in recent decades, the artificial joint is a place where frictional motion continues to occur and is permanently unavailable due to the generation of the wear particles and dissociation of the insert [4]. It can also lead to complications by biological reactions such as inflammation caused by FPs [40]. There are various materials for artificial joints, including metal and polyethylene. Alumina ceramic has attracted the attention due to its high chemical, physical and biological stability. However, inflammation has been often seen around the implant tissue despite being an artificial joint using Al_2_O_3_ [3]. Moreover, tissue fibrosis at the periprosthetic joint capsule collected during a revision surgery was significantly increased than in the metal and polyethylene articulation due to inflammatory response [5].

It has been shown that the Al_2_O_3_ induces the activation of human macrophages [2,3,40]. In addition, we further found that the systemic injection of Al_2_O_3_ FPs promoted the activation of macrophages, Th1, and Tc1 cells, leading to exacerbation of the inflammatory diseases such as colitis. Therefore, these data are the first report demonstrating the possibility of inflammation exacerbation by Al_2_O_3_ FPs in the human.

Macrophages contain two phenotypes: M1 and M2 [41,42]. M1 macrophages contribute in the clearance of microbes as inflammatory immune responses, especially the production of pro-inflammatory cytokines and upregulation of co-stimulators and MHC molecules [43]. In contrast to the M1 phenotype, the M2 macrophages contribute to the repairing of the inflamed tissue by suppressing immune activation [44]. Therefore, M1 and M2 macrophages are competitive in immune responses. In this study, we found that the treatment with the Al_2_O_3_ FPs promoted upregulation of co-stimulators and production of pro-inflammatory cytokines in the macrophages, indicating the activation of the M1 macrophages. Thus, Al_2_O_3_ FPs contribute to inflammation of the peripheral tissue instead of M2-mediated tissue-repair.

DSS is one of the materials that induces colitis in the mouse model [45,46]. It causes the loss of the colon’s epithelial barrier function and induces infiltration of the immune cells in the colon, which induces the inflammation of the tissues [36,45]. The DSS-induced colitis is mediated by the Th1 and Th17 cells [47,48]. It is well known that the continuous administration of the DSS, such as 7 days administration, promotes serious inflammation of the colon [49]. We administrated the DSS for 5 days and followed by the treatment with the distilled water or Al_2_O_3_ FPs for the determination of the effect of Al_2_O_3_ FPs on the colitis. The 5-day administration of DSS did not promote serious inflammation of the colon, and the exacerbation effect of Al_2_O_3_ FPs in the mice was evaluated. Additional treatment with the Al_2_O_3_ FPs in the DSS-induced colitis in the mice further upregulated the infiltration of T cells and pro-inflammatory cytokines. Thus, these data demonstrated the exacerbation of inflammatory diseases by the Al_2_O_3_ FPs.

Treatment with the Al_2_O_3_ FPs induced the Th1 and Tc1 responses in the mice. In the DSS-colitis models, the additional treatment with the Al_2_O_3_ FPs further increased the Th1 and Tc1 cells in the colon. Interestingly, IL-17-producing CD4 T cells were also increased in the DSS-induced colitis mice by Al_2_O_3_ FPs; however, Al_2_O_3_ FPs treatment alone did not induce Th17 responses in the mice in vivo. Such different effects on the induction of the Th17 response may be due to the synergistic effect of DSS and Al_2_O_3_ FPs. Although the Al_2_O_3_ FPs did not promote significant difference in the induction of Th17 responses, treatment with Al_2_O_3_ FPs after DSS treatment might increase the Th17 response synergistically. In our next study, we will evaluate whether the DSS-induced increases in the levels of RORγδ, a transcription factor of Th17 cells [50], are further increased by the Al_2_O_3_ FPs.

## 4. Materials and Methods

### 4.1. Mice

C57BL/6 mice were purchased from Hyochang Science (Daegu, Korea) and used at the age of 5 to 8 weeks. The mice were kept under pathogen-free standard conditions in a room with sterilized chow and water, 20–25 °C, and 40–60% humidity. The study was approved by the Yeungnam University Institutional Animal Care and Use Committee (IACUC) (21 July 2020) of Yeungnam University, Korea, and it is recorded recorded by the animal protocol number 2020-011, in compliance with the Animal Protection Act, the Law on Laboratory Animals, and the IACUC regulation of Yeungnam University. All the experiments in this study were performed with the efforts to minimize the pain to the animals, and CO_2_ gas was used for euthanasia in accordance with the humanitarian end-point standards in terms of ethics.

### 4.2. Reagents

Al_2_O_3_ FPs were purchased from Thermo Fisher Scientific (Waltham, MA, USA, #06300). Al_2_O_3_ FPs were prepared in suspension in PBS (Thermo Fisher Scientific, Waltham, MA, USA, #10010023). LPS from *Escherichia coli* (O111:B4 type, #L2630) was purchased from Sigma-Aldrich (St. Louis, MO, USA). DSS (40 kDa, Sigma-Aldrich, St. Louis, MO, USA, #42867) was dissolved at 5% concentration in sterile drinking water, placed in cage water bottles, and provided to mice.

### 4.3. Transmission Electron Microscopy (TEM)

The size and shape of Al_2_O_3_ FPs were determined by TEM (H-7600 transmission electron microscope, Hitachi, Japan). Sample was prepared by making suspension of Al_2_O_3_ FPs in distilled water (DW), followed by deposition onto the TEM grid.

### 4.4. Antibodies

Isotype control antibodies such as IgG1, IgG2a and IgG2b, Fc-block antibodies, APC-anti-CD317/BST2 (927, #127015), BV785-anti-B220/CD45R (RA3-6B2, #103245), APC-Cy7-anti-CD11c (N418, #117323), BV785-anti-CD11c (N418, #117335), PerCp5.5-anti-CD11b (M1/70, #101227), BV510-anti-NK1.1 (PK136, #108737), FITC-anti-CD3 (17A2, #100203), FITC-anti-CD90.1/Thy-1.1 (OX-7, #202503), FITC-anti-Ly-6G/Ly-6C/Gr-1 (RB6-8C5, #108405), FITC-anti-CD49b (HMα2, #103503), FITC-anti-TER-119 (TER-119, #116205), APC-anti-CD40 (3/23, #124611), FITC-anti-CD40 (3/23, #124607), BV605-anti-CD80 (16-10A1, #104729), PE-Cy7-anti-CD86 (GL-1, #105013), APC-Cy7-anti-CD8α (53-6.7, #100713), PerCp5.5-anti-CD4 (GK1.5, #100433), PE-anti-CD178/Fas ligand (MFL3, #106605), APC-anti-CD253/TRAIL (N2B2, #109309), APC-Cy7-anti-CD69 (H1.2F3, #104525), Alexa Fluor 647-anti-Granzyme B (GB11, #515405), PE-anti-Perforin (S16009A, #154305) PerCp5.5-anti-MHC class I/H-2Kb (28-8-6, #114619), PE-anti-MHC class II/I-A/I-E (M5/114.15.2, #107607), PE-Cy7-anti-IFN-γ (XMG1.2, #505825), PerCp5.5-anti-IL-4 (11B11, #504123), PE-IL-17A (TC11-18H10.1, #506903), and APC-anti-TNF-α (MP6-XT22, #506307) antibodies were obtained from BioLegend (San Diego, CA, USA).

### 4.5. Preparation of Single Cell Suspension from Tissues

The spleens, colon, and mLNs were harvested and digested with digestion buffer containing DNase I (Sigma-Aldrich, MO, USA, #DN25) and collagenase D (Sigma-Aldrich, MO, USA, #11088858001) for 20 min at room temperature (RT). After washing with PBS, the pellets were re-suspended in 5 mL of histopaque 1.077 (Sigma-Aldrich, MO, USA, #10771), forming a layer on the fresh 5 mL histopaque, followed by fresh media on the top. The cells were centrifuged for 10 min at 1836× *g*. The leukocytes-containing fraction, with a density less than 1.077 g/mL, was collected [51].

### 4.6. Analysis of Innate Immune Cells

The single cells were stained with unlabeled isotype control antibodies and Fc-block antibodies to block non-specific binding before staining with fluorescence-conjugated antibodies. For the analysis of pDCs, the splenocytes were stained with APC-anti-CD317/BST2, BV785-anti-B220/CD45R, and APC-Cy7-anti-CD11c antibodies, and CD11c^-^B220^+^CD317^+^ cells were defined in the live leukocytes. Lineage staining and CD11c were used for defining cDCs. FITC-anti-CD3, FITC-anti-CD90.1, FITC-anti-Gr-1, FITC-anti-CD49b, and FITC-anti-TER-119 were used for lineage staining. The cells of CD11c^+^ lineage^-^ in the live leukocytes were defined as cDCs [26]. The macrophages were defined as CD11c^inter^CD11b^+^ cells in the splenocytes. NK cells were stained with BV510-anti-NK1.1 and FITC-anti-CD3 and defined as CD3^-^NK1.1^+^ cells in the splenocytes [52]. The dead cells were removed during the flow cytometric analysis by staining with 4′,6-diamidino-2-phenylindole solution (DAPI, Sigma-Aldrich, MO, USA, #D9542).

### 4.7. Stimulation and Intracellular Staining of T cells

The single cells were incubated with ionomycin (1 Μm, #I9657) and phorbol 12-myristate 13-acetate (PMA, 50 ng/mL, #P8139) for 2 h (Sigma-Aldrich, St. Louis, MO, USA). The cells were further incubated with monensin solution (2 μM; BioLegend, San Diego, CA, USA, #420701) for 2 h. The cells were stained with surface antibodies; T cells with FITC-anti-CD3, BV510-anti-NK1.1, PerCp5.5-anti-CD4, and APC-Cy7-anti-CD8α and NK cells with FITC-anti-CD3 and BV510-anti-NK1.1. The cells were washed with PBS, fixed, and permeabilized by Cytofix/Cytoperm buffer (eBioscience, San Diego, CA, USA, #88-8824-00). After washing with Perm/Wash buffer (eBioscience, San Diego, CA, USA, #00-8333-56), the cells were stained with intracellular antibodies in Perm/Wash buffer; T cells with PE-Cy7-anti-IFN-γ, PerCp5.5-anti-IL-4, PE-anti-IL-17A, and APC-anti-TNF-α and NK cells with APC-anti-Granzyme B, PE-anti-Perforin, and PE-Cy7-anti-IFN-γ at RT for 30 min. The dead cells were isolated using Zombie Violet Fixable Viability Kit (BioLegend, San Diego, CA, USA, #423113). This experiment pre-stained unlabeled isotype control antibodies. Fc-blocking antibodies were used as negative controls in all the experiments in this study. The cells were analyzed by NovoCyte Flow Cytometer and NovoExpress Software (ACEA Biosciences, San Diego, CA, USA).

### 4.8. Enzyme Linked Immunosorbent Assay (ELISA)

Concentrations of IFN-γ, Interleukin-6 (IL-6, #431301), IL-12p70 (#433604), and TNF-α (#430901) in the sera were measured by ELISA kits (BioLegned San Diego, CA, USA) in triplicates.

### 4.9. Mouse Model of DSS-Induced Colitis

The mice were randomly divided into 4 groups (5 mice in each group). The first group was a control group provided with normal water daily. In the second group, in addition to providing normal water daily, Al_2_O_3_ FP (10 mg/kg) was *i.p*. injected from day 6 to 8. The 3rd and 4th groups were provided with 5% DSS in the water for 5 days. From day 6, the 3rd group was provided with normal water and daily injection of PBS whereas the 4th group was provided with normal water and injection of Al_2_O_3_ FPs for 3 days.

### 4.10. Histological Analysis

The small fractions of colon in each group were fixed in 4% formaldehyde. The colons were dehydrated overnight with 50%, 70%, 95%, and 100% ethanol. The colons were then embedded in paraffin. The paraffin blocks were cut into 6-mm-thick sections, placed on glass slides, and dried at 50–55 °C on a hot plate. The sections were deparaffinized with xylene, rehydrated, and then stained with H&E (Sigma-Aldrich, MO, USA, #H3136 and #E4009). These sections were dehydrated, covered with a cover glass with Canada balsam (Sigma-Aldrich, MO, USA, #C1795), and observed under a microscope.

### 4.11. Statistical Analyses

All the data are shown as mean ± standard error of the mean (SEM). Data sets were analyzed using a two-way ANOVA (Tukey multiple comparison test) and Mann–Whitney *t*-test. *p* values < 0.05 were considered to have a statistically significant difference. *p* < 0.05 was expressed as “*”.and *p* < 0.01 was expresses as “**”.

## 5. Conclusions

In this study, we demonstrated that Al_2_O_3_ FPs induces the activation of macrophages, which elicits differentiation and activation of CD4 T and CD8 T cells. In addition, Al_2_O_3_ FPs promoted inflammatory immune response progression of DSS-induced colitis in the mice. Thus, these data suggested that Al_2_O_3_ FPs produced from ceramic may be able to contribute to the induction and exacerbation of inflammatory diseases in animals and humans.

## Figures and Tables

**Figure 1 ijms-21-07114-f001:**
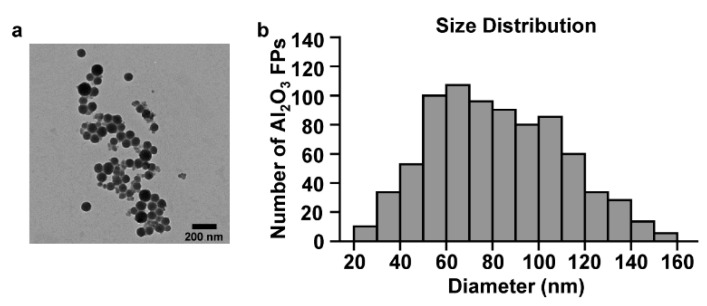
Characteristics of the aluminum oxide fine particles (Al_2_O_3_ FPs). (**a**) Transmission electron micrograph, (**b**) size distribution of Al_2_O_3_ FPs.

**Figure 2 ijms-21-07114-f002:**
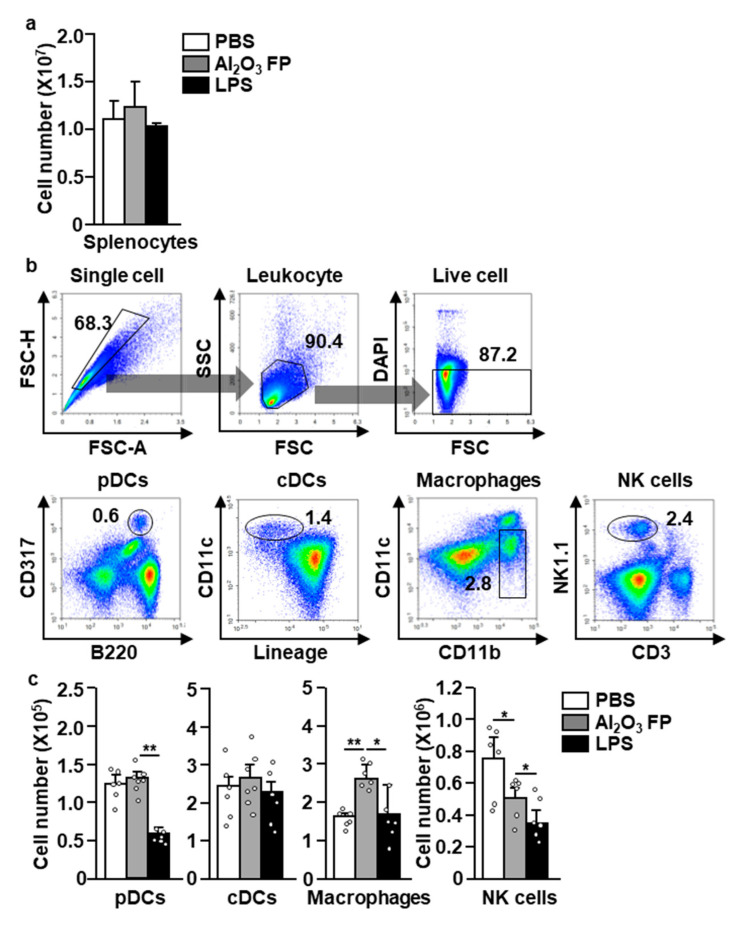
The effect of Al_2_O_3_ FPs on the innate immune cells. The C57BL/6 mice were intraperitoneally (*i.p*.) injected with 10 mg/kg of Al_2_O_3_ FPs. Six hours post-injection, the spleens were harvested and the splenocytes were analyzed. (**a**) Total numbers of splenocytes. (**b**) Definition of plasmacytoid dendritic cells (pDCs), conventional dendritic cells (cDCs), macrophages and natural killer (NK) cells. (**c**) The absolute numbers of pDCs, cDCs, macrophages and NK cells analyzed in the spleen. All data are representatives and averages of six independent samples (2 mice for three experiments, two-way ANOVA, mean ± SEM). * and ** represent *p* < 0.05 and *p* < 0.01, respectively.

**Figure 3 ijms-21-07114-f003:**
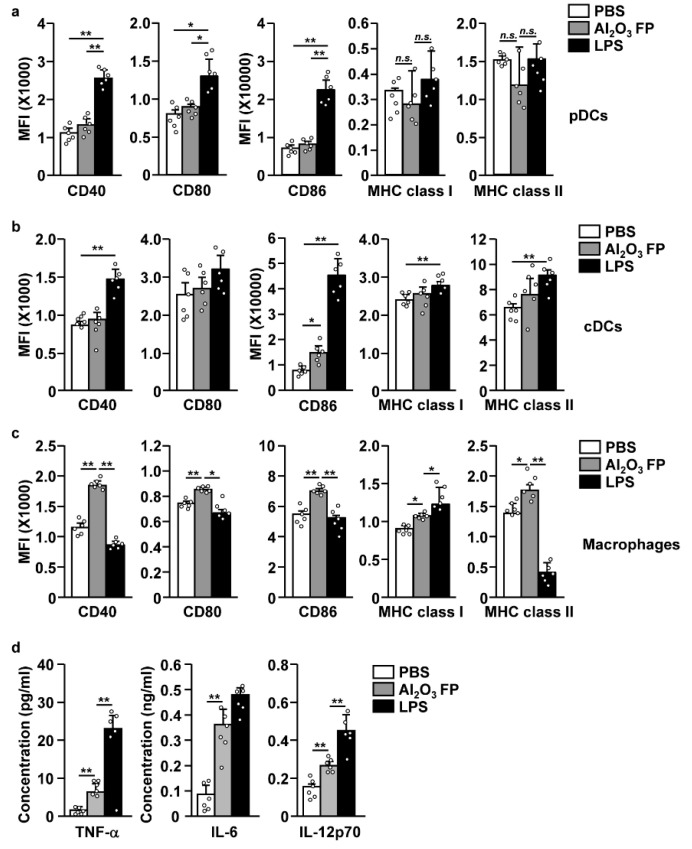
Al_2_O_3_ FPs promoted the activation of macrophages, but not pDCs and cDCs. The C57BL/6 mice were *i.p.* injected with 10 mg/kg of Al_2_O_3_ FPs. Spleens were harvested six hours post-injection, followed by flow cytometric analysis of the pDCs, cDCs and macrophages. The expression levels of co-stimulator and major histocompatibility complex (MHC) class I and II were shown in (**a**) pDCs, (**b**) cDCs and (**c**) macrophages (*n* = 6 mice, two-way ANOVA, mean ± SEM). (**d**) Concentration of tumor necrosis factor-alpha (TNF-α), interleukin (IL)-6, and IL-12p70 were measured by Enzyme Linked Immunosorbent Assay (ELISA). The data are averages of six independent samples (2 mice for three experiments, two-way ANOVA, mean ± SEM). * and ** represent *p* < 0.05 and *p* < 0.01, respectively.

**Figure 4 ijms-21-07114-f004:**
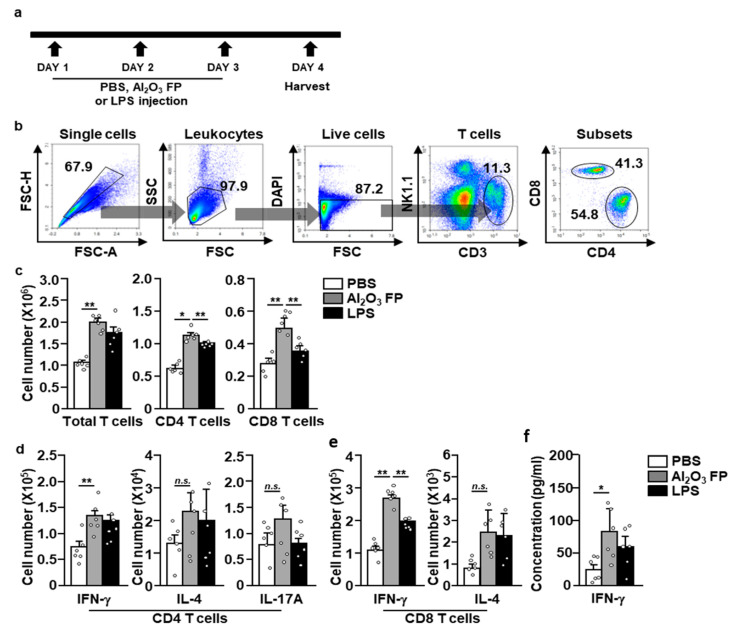
The treatment with Al_2_O_3_ FPs promoted the activation of T cells. The C57BL/6 mice were treated with 10 mg/kg of Al_2_O_3_ FPs for 3 days. On day 4, the mice were sacrificed, and their spleens were harvested. (**a**) Schematic diagram of T cell activation experiment. (**b**) Definition of spleen T cells. (**c**) The number of total T cells, CD4 T cells, and CD8 T cells in the spleen. The number of intracellular cytokine producing CD4 T cells and CD8 T cells in the spleen (**d**) and (**e**). (**f**) The serum concentration of interferon-gamma (IFN-γ) analyzed by ELISA. All data are representatives and averages of six independent samples (2 mice for three experiments, two-way ANOVA, mean ± SEM). * and ** represent *p* < 0.05 and *p* < 0.01, respectively.

**Figure 5 ijms-21-07114-f005:**
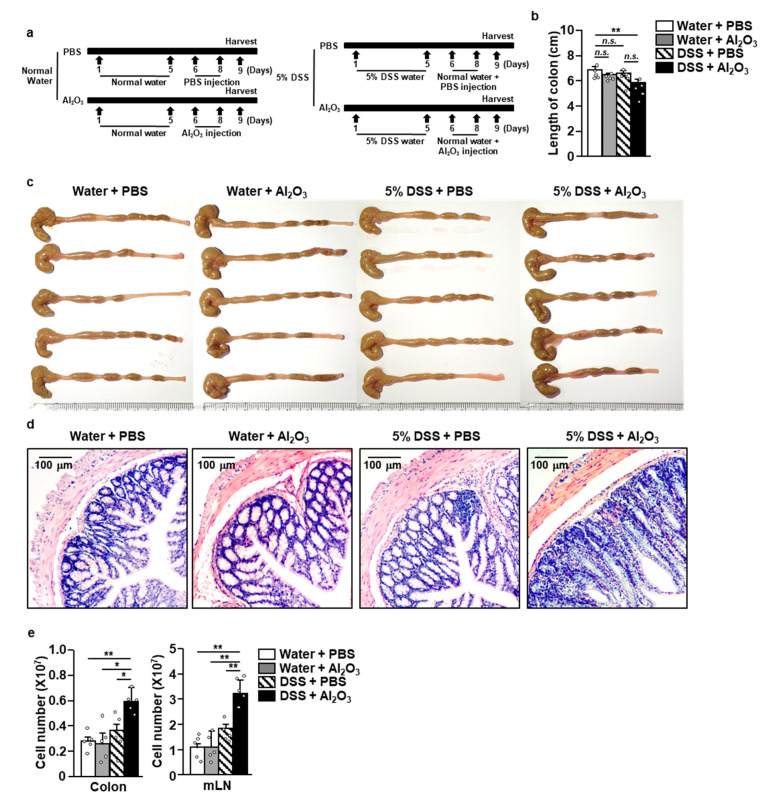
Al_2_O_3_ FPs exacerbated the dextran sulfate sodium (DSS)-induced colitis. C57BL/6 mice developed colitis as shown in the method section. (**a**) Schematic diagram of four groups: Water + phosphate buffered saline (PBS) (*n* = 5), Water + Al_2_O_3_ FPs (*n* = 5), DSS + PBS (*n* = 5), and DSS + Al_2_O_3_ FPs (*n* = 5). On day 9, the mice were sacrificed and their colon and mesenteric lymph nodes (mLNs) were harvested. (**b**) Length of the colons. (**c**) The colons. (**d**) Hematoxylin and eosin (H&E) staining of the colons. (**e**) The numbers of colon-infiltrated leukocytes and the numbers of mLN cells (*n* = 10, 5 mice for two experiments, two-way ANOVA, mean ± SEM). * and ** represent *p* < 0.05 and *p* < 0.01, respectively.

**Figure 6 ijms-21-07114-f006:**
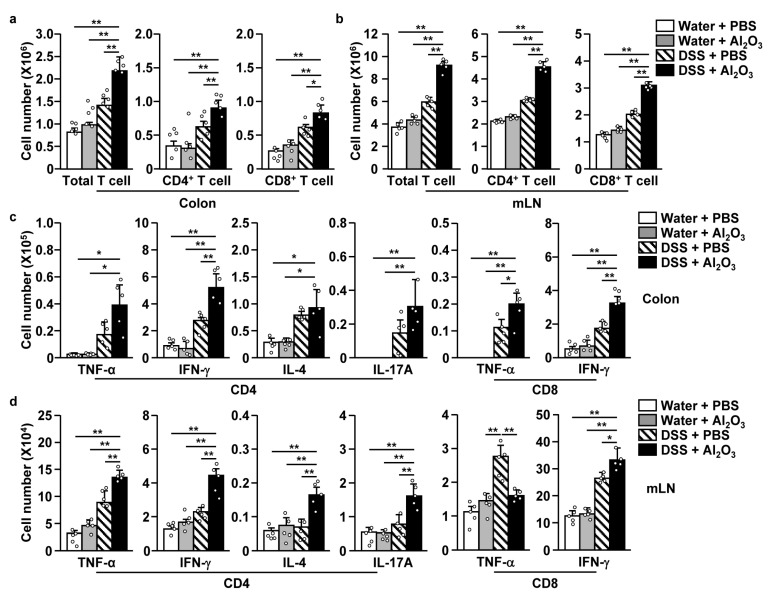
Al_2_O_3_ FPs contributed in the T cell-mediated inflammatory responses. The C57BL/6 mice were treated with the DSS and Al_2_O_3_ FPs as shown in Figure 5. (**a**) Total numbers of colon infiltrated T cells, CD4 T cells and CD8 T cells. (**b**) The number of total T cells, CD4 T cells and CD8 T cells in mLNs. (**c**) The numbers of cytokine producing CD4 and CD8 T cells in colon. (**d**) Intracellular cytokine-producing CD4 and CD8 T cells in the mLNs (*n* = 5, two-way ANOVA, mean ± SEM). * and ** represent *p* < 0.05 and *p* < 0.01, respectively.

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
