# Peer review of "Alumina Ceramic Exacerbates the Inflammatory Disease by Activation of Macrophages and T Cells"

_ijms, 2020, doi:10.3390/ijms21197114_

Round 1
Reviewer 1 Report
The topic could be of interest but serious flaws concerning the rationale of the research are present. In particular:
- The Authors asses that aluminum oxide (Al2O3) ceramic is one of the materials used for artificial joints but then they inject the product intraperitoneally and also use as experimental model IBD: which is the link whit the possible particles release from artificial joints?
- The authors use as positive control LPS, why? Why they think LPS is appropriated?
- Why the authors used the dose of 10mg/kg? There is a relationship with the particles released from joints? These levels are intraperitoneally accumulated in patients?
Author Response
Reviewer #1
The topic could be of interest but serious flaws concerning the rationale of the research are present. In particular:
- The Authors asses that aluminum oxide (Al2O3) ceramic is one of the materials used for artificial joints but then they inject the product intraperitoneally and also use as experimental model IBD: which is the link whit the possible particles release from artificial joints?
Answer (A): After implantation of an artificial joint, fine particles (FPs) are produced by wear, which are observed around the tissues at the implant site, but also those are spread throughout the peripheral tissues along the blood vessels. This study confirmed whether those FPs, when spread throughout the peripheral tissues, affect the degree of disease progression in patients with inflammatory conditions. There are no reports of the relationship between inflammatory disease and FPs and risk of FPs of Al2O3 to inflammation. Therefore, this study will be first report that the Al2O3 FPs may be contribute progression of inflammatory diseases.
- The authors use as positive control LPS, why? Why they think LPS is appropriated?
A: In this study, we evaluated the possibility of inducing inflammatory response of Al2O3 fine particles (FPs) in association with immune cells. Because the LPS is the most well studied molecule for induction of immune cell activation and inflammatory immune responses, we compared the effect of Al2O3 FPs in induction of immune cell activation with LPS.
- Why the authors used the dose of 10 mg/kg? There is a relationship with the particles released from joints? These levels are intraperitoneally accumulated in patients?
A: We have examined dose-dependent effect of Al2O3 FPs in the induction of immune cell activation, but we skipped to add result for summarizing the data. We have now revised and added the dose- and time-dependent effect of Al2O3 FPs at Figure S1 and S2. As shown in the Figure S1, 10 mg/kg of Al2O3 FPs promoted substantial upregulation of co-stimulators on macrophages. We also revised main text with appropriated explanation of the data. Moreover, although there is evidence that Al2O3 FPs were produced from artificial joint, the concentration of the Al2O3 FPs is not able to determine, because it easily spread out to peripheral tissues.
Reviewer 2 Report
Summary:
In the manuscript “Alumina Ceramic Exacerbates the Inflammatory Disease by Activation of Macrophages and T cells” Seong-Min Lim et al. describe their analysis of the effect of FWPs of Al2O3 on various immune cells, including T cells, macrophages and DCs. They found that these compounds can increase the number of macrophages and T cells as well as various activation markers in macrophages and T cells. The effect of FWPs of Al2O3 were also examined in a mouse model of colitis.
Points to address:
- In the introduction, the shift to a paragraph about IBD was pretty abrupt. Is there a specific link between people with an artificial joint/hip and IBD or other GI-related disorders? Or is the reason for IBD simply because it is a model for inflammation?
- For Figure 1, what was the number of particles examined in 1b? Is this size distribution representative of what we know about the fine wear particles in vivo in humans?
- Is there a particular reason why a concentration of 1mg/kg of Al2O3 FWPs was used? Was a range of doses used? If there is previous literature citing a precedent for this, please cite it. If not, please give reasoning for why this dose was appropriate.
- For Figure 5, the text says histological analysis was performed but I do not see any quantification of the histological analysis, just images. Was this quantified? For the IBD model, was PBS used as the vehicle for the control or just water vs. DSS? Was this mouse experiment repeated or only performed 1 time? Even though there were 5 mice per group, the experiment should really be performed at least twice.
Author Response
Reviewer #2
In the manuscript “Alumina Ceramic Exacerbates the Inflammatory Disease by Activation of Macrophages and T cells” Seong-Min Lim et al. describe their analysis of the effect of FWPs of Al2O3 on various immune cells, including T cells, macrophages and DCs. They found that these compounds can increase the number of macrophages and T cells as well as various activation markers in macrophages and T cells. The effect of FWPs of Al2O3 were also examined in a mouse model of colitis.
Points to address:
- In the introduction, the shift to a paragraph about IBD was pretty abrupt. Is there a specific link between people with an artificial joint/hip and IBD or other GI-related disorders? Or is the reason for IBD simply because it is a model for inflammation?
Answer (A): Thank you for the reviewer’s valuable comments. It has shown that Al2O3 FPs promote immune activation, but the relation of inflammatory disease and Al2O3 FPs has not investigated. Therefore, there is still no reports that Al2O3 FPs contribute development and exacerbation of inflammatory diseases. Based on previous study that Al2O3 FPs induce activation of macrophages, we further examined the effect of Al2O3 FPs in mouse in vivo and contribution of disease development. Honestly, the DSS-induced colitis is the easiest mouse model, which is why we examined the effect of Al2O3
- For Figure 1, what was the number of particles examined in 1b? Is this size distribution representative of what we know about the fine wear particles in vivo in humans?
A: Unfortunately, It is impossible to harvest Al2O3 fine wear particles from patient. Thus, the size and amount of the Al2O3 FPs were not able to determine. We changed the frequency in Figure 1b to the actual counting number. - Is there a particular reason why a concentration of 10mg/kg of Al2O3 FWPs was used? Was a range of doses used? If there is previous literature citing a precedent for this, please cite it. If not, please give reasoning for why this dose was appropriate.
A: We have examined dose- and time-dependent effect of Al2O3 FPs in induction of immune cell activation. We have now added the data in Figure S1 and S2. Since 10 mg/kg of Al2O3 FPs showed dramatic increases in the costimulatory expression in macrophages, we evaluated 10 mg/kg of Al2O3 FPs in the contribution of inflammatory disease in the mouse.
- For Figure 5, the text says histological analysis was performed but I do not see any quantification of the histological analysis, just images. Was this quantified? For the IBD model, was PBS used as the vehicle for the control or just water vs. DSS? Was this mouse experiment repeated or only performed 1 time? Even though there were 5 mice per group, the experiment should really be performed at least twice.
A: We are sorry for making confuse you. We have now revised the histology images. Based on experiment plan, the mice received DSS water for 5 days and further treated with PBS or 10 mg/kg of Al2O3 FPs for additional 3 days. The PBS treatment was used as a control. In addition, we repeated the mouse experiment and revised the result in the figure. The data were reproducible.
Reviewer 3 Report
Alumina ceramic exacerbates the inflammatory diseases by activation of macrophages and T cells by Lim et al.
This manuscript demonstrates the role of Al2O3 FWPs on inflammation in mice. Intraperitonel administration of FWPs induced increase number of macrophages in spleen. In addition, these macrophages showed up-regulation of several activation markers such as CD40 and MHC class II. Not only innate cells, Al2O3 FWPs promote the activation of Th1 and Tc1 cells. In the case of colitis model induced by DSS, Al2O3 FWPs induced infiltration of inflammatory cells into colon, activation of Th1, Tc1 and Th17, and exacerbated colon inflammation, as a result. These findings were of interest and clinically important, however following points should be addressed to improve this manuscript.
- The authors used Al2O3 FWPs in this study, but is this particle actual `wear’ particle? In materials and methods section, the authors only showed that this reagent was purchased from Thermo, but I was not able to find this `wear’ particle as a product by Web site. The authors should describe the information of this reagent in detail. If the authors used normal fine particles but not actual `wear’ particle, the authors should describe that Al2O3 fine particles are used as model of Al2O3 fine wear particles. In addition, the authors should change FWPs to fine particles (FPs). Readers will misread that.
- Did the authors examine endotoxin levels of Al2O3 FWPs? In some cases, endotoxin contamination is observed in purchased product.
- In materials and methods section, other information is also lacked.
LPS: Which LPS did you use?
DSS: Which DSS did you use? Different molecular weight DSS have different characteristics for the induction of DSS colitis.
In any case, the authors should describe product number of reagents (also all reagent in this study). - In figure2 and 3, all data is from six hours after injection of FWPs. Why did you analyze immune responses at six hours? Do you have any evidences the reason why 6 hours is good for analysis of the effect of FWPs?
- The authors used 10 mg/kg of Al2O3 FWPs for in vivo experiment. It estimates 200 microg/mouse. How did you find an optimal dose of FWPs?
- ? After injection of Al2O3 FWPs, activation of Th1 and Tc1 (and Th17 in DSS colitis model) were observed. In general, activation of T cells are antigen-specific responses. How do you think about that? Are these T cells responses antigen-specific or polyclonal activation?
- In figure 5b and 5d, pictures should be improved. In page 5 line146-150, the authors explained regarding rectal bleeding and infiltration of cells in the colon. However, it is difficult to understand the authors claim from these pictures. High resolution and enlarged views are required in this figure.
Author Response
Reviewer #3
Alumina ceramic exacerbates the inflammatory diseases by activation of macrophages and T cells by Lim et al.
This manuscript demonstrates the role of Al2O3 FWPs on inflammation in mice. Intraperitoneal administration of FWPs induced increase number of macrophages in spleen. In addition, these macrophages showed up-regulation of several activation markers such as CD40 and MHC class II. Not only innate cells, Al2O3 FWPs promote the activation of Th1 and Tc1 cells. In the case of colitis model induced by DSS, Al2O3 FWPs induced infiltration of inflammatory cells into colon, activation of Th1, Tc1 and Th17, and exacerbated colon inflammation, as a result. These findings were of interest and clinically important, however following points should be addressed to improve this manuscript.
- The authors used Al2O3 FWPs in this study, but is this particle actual `wear’ particle? In materials and methods section, the authors only showed that this reagent was purchased from Thermo, but I was not able to find this `wear’ particle as a product by Web site. The authors should describe the information of this reagent in detail. If the authors used normal fine particles but not actual `wear’ particle, the authors should describe that Al2O3 fine particles are used as model of Al2O3 fine wear particles. In addition, the authors should change FWPs to fine particles (FPs). Readers will misread that.
Answer (A): Thank for the comment. We agreed the reviewer’s comment and we have now changed fine wear particles (FWPs) to fine particles (FPs). - Did the authors examine endotoxin levels of Al2O3 FWPs? In some cases, endotoxin contamination is observed in purchased product.
A: The Al2O3 FPs was purchased from Sigma-Aldrich, which was 99.7+% pure product. We also measured the endotoxin levels and showed lower than 0.01 EU/ml. - In materials and methods section, other information is also lacked.
LPS: Which LPS did you use?
DSS: Which DSS did you use? Different molecular weight DSS have different characteristics for the induction of DSS colitis.
In any case, the authors should describe product number of reagents (also all reagent in this study).
A: We used 0111:B4 type LPS and 40kDa DSS for the experiment. We have now revised material and method section. - In figure2 and 3, all data is from six hours after injection of FWPs. Why did you analyze immune responses at six hours? Do you have any evidences the reason why 6 hours is good for analysis of the effect of FWPs?
A: We have examined the effect of Al2O3 FPs in innate immune cells 6 and 18 hours after treatment (Figure S2). Six hours after injection of the Al2O3 FPs, co-stimulators were significantly increased in macrophages, which were rapidly decreased 18 hours after injection of Al2O3 FPs. Thus, we showed the six hour data in Figure 2. Consistent with previous study, the immune cells were activated quite early time point in the mouse in vivo compared in vitro assay (reviewer references [1]). - The authors used 10 mg/kg of Al2O3 FWPs for in vivo experiment. It estimates 200 microg/mouse. How did you find an optimal dose of FWPs?
A: The dose of Al2O3 FPs has determined by evaluation of dose-dependent effect of Al2O3 FPs in the mouse in vivo. We have now added the dose-dependent data in Figure S1.
- After injection of Al2O3 FWPs, activation of Th1 and Tc1 (and Th17 in DSS colitis model) were observed. In general, activation of T cells are antigen-specific responses. How do you think about that? Are these T cells responses antigen-specific or polyclonal activation?
A: As the reviewer mentioned, the activation of T cells was controlled by dendritic cells (DCs). The DC presented antigen, co-stimulator and cytokine will promote differentiation of naïve T cells to effect T cells. In case of autoimmune disease, there is still no clear evidence for antigen, whether the activated T cells are antigen-specific or polyclonal. Moreover, in vitro T cell stimulation is simply conducted without any antigenic protein, which the naïve T cells stimulated with anti-CD3 and anti-CD28 antibodies together with appropriate cytokines. Therefore, the Al2O3 FPs promoted development of colitis may be the polyclonal T cell responses.
- In figure 5b and 5d, pictures should be improved. In page 5 line146-150, the authors explained regarding rectal bleeding and infiltration of cells in the colon. However, it is difficult to understand the authors claim from these pictures. High resolution and enlarged views are required in this figure.
A: We are sorry for the poor image. We have repeated the experiment again and revised the image.
Reviewer Reference
[1] L. Xu, M. Kwak, W. Zhang, P.C. Lee, J.O. Jin, Time-dependent effect of E. coli LPS in spleen DC activation in vivo: Alteration of numbers, expression of co-stimulatory molecules, production of pro-inflammatory cytokines, and presentation of antigens, Mol Immunol 85 (2017) 205-213.
Round 2
Reviewer 1 Report
The revised version is partially improved but the reply to the first question is not satisfactory.
There are evidences, also in literature, that these particles accumulate at intraperitoneal level? Then, how the intraperitoneal accumulation can be linked to the intestinal effects? How, from peritoneum, the particles can reach the intestinal lumen?
Reviewer 2 Report
They addressed my concerns.
Reviewer 3 Report
All revised data are persuasive, and author's responses are satisfied me.